# Research on short-term power load forecasting based on VMD and GRU

**Haoyue Sun, Zhicheng Yu\*, Bining Zhang**

College of Information Engineering, Hebei University of Architecture, Zhangjiakou, China

\* 306561478@qq.com

**Data Availability Statement:** All relevant data can be found at Github: https://github.com/yzc-dot/Electricity-prediction.

**Funding:** The author(s) received no specific funding for this work.

## Abstract

The traditional method for power load forecasting is susceptible to various factors, including holidays, seasonal variations, weather conditions, and more. These factors make it challenging to ensure the accuracy of forecasting results. Additionally, there is a limitation in extracting meaningful physical signs from power data, which ultimately reduces prediction accuracy. This paper aims to address these issues by introducing a novel approach called VCAG (Variable Mode Decomposition—Convolutional Neural Network—Attention Mechanism—Gated Recurrent Unit) for combined power load forecasting. In this approach, we integrate Variable Mode Decomposition (VMD) with Convolutional Neural Network (CNN). VMD is employed to decompose power load data, extracting valuable time-frequency features from each component. These features then serve as input for the CNN. Subsequently, an attention mechanism is applied to give importance to specific features generated by the CNN, enhancing the weight of crucial information. Finally, the weighted features are fed into a Gated Recurrent Unit (GRU) network for time series modeling, ultimately yielding accurate load forecasting results.To validate the effectiveness of our proposed model, we conducted experiments using two publicly available datasets. The results of these experiments demonstrate that our VCAG method achieves high accuracy and stability in power load forecasting, effectively overcoming the limitations associated with traditional forecasting techniques. As a result, this approach holds significant promise for broad applications in the field of power load forecasting.

## Introduction

At present, there is a growing global demand for electricity, with global electricity demand returning to growth for two consecutive years after a slight decline in 2020. In 2022, global electricity demand increased by approximately 2% year-on-year, which is in line with the average growth rate during the five years preceding the COVID-19 outbreak (2015–2019) at 2.4%, but significantly lower than the growth rate in 2021, which was 6%. According to relevant data, global electricity demand is expected to rise by more than 30% from 2020 to 2030.

The International Energy Agency has noted that extreme weather events in various regions of the world in 2022 have highlighted the need to enhance the security of electricity supply.

**Competing interests:** The authors have declared that no competing interests exist.

Factors such as weather conditions, equipment failures, and emergencies can impact the supply and quality of electricity, leading to imbalances between power demand and supply. Therefore, accurate power forecasting has become a crucial challenge for power enterprises.

Initially, statistical methods dominated power load forecasting [1]. The Autoregressive Moving Average (ARIMA) model was introduced in the 1970s for time series analysis to capture trends, seasonality, and noise in electricity load data [2, 3]. Subsequently, generalized autoregressive conditional heteroscedastic (GARCH) models and other extended models were developed to address nonlinear problems. Smoothing methods like exponential smoothing and cubic smoothing gained traction for load data smoothing and noise removal. Regression analysis was used to establish relationships between external factors such as meteorological and economic data and power load, with multiple linear regression and nonlinear regression models gradually introduced.

While statistical methods were widely used initially, they had limitations in dealing with nonlinear relationships and long-term dependencies. They also required stable data and fixed model parameters, making them less adaptable to dynamic data changes.

The introduction of machine learning methods addressed these limitations. With improved computing power and access to large-scale datasets, machine learning became increasingly relevant in power load forecasting. Support Vector Machines (SVMs) helped tackle nonlinear problems by finding optimal hyperplanes for nonlinear regression [4]. Random Forest, as an ensemble learning algorithm, combined multiple decision tree models to enhance prediction stability and performance. The k-nearest Neighbor (K-NN) method used distance metrics for simple load forecasting. Decision tree models gained popularity due to their interpretability and ease of understanding.

Machine learning methods saw significant progress, but they still required human intervention in feature engineering, and model performance depended heavily on feature selection and parameter tuning. Moreover, machine learning methods often demand large datasets for accurate predictions, which could be a limitation in some power load forecasting scenarios.

The emergence of deep learning methods addressed many of these issues. Deep learning methods rose in prominence thanks to the development of neural networks and improved computing power [5]. While Recurrent Neural Networks (RNNs) were initially used for sequence data modeling, their performance on long sequences was limited due to the vanishing gradient problem. The Long Short-Term Memory (LSTM) network, an improved variant of RNN, overcame the vanishing gradient problem and excelled in capturing long-term dependencies [6]. Convolutional Neural Networks (CNNs), originally designed for image processing, found application in power load data for feature extraction through convolution and pooling operations [7]. Deep Neural Networks (DNNs) were employed for nonlinear modeling and high-dimensional data, utilizing multiple hidden layers for feature extraction and load forecasting.

Deep learning methods delivered impressive results in power load forecasting, but they demanded significant computational resources and training time, which could be impractical in resource-constrained applications. Additionally, deep learning models were often complex and challenging to interpret, which might not be desirable in certain scenarios.

In this paper, we introduce a comprehensive model known as VCAG. VCAG is a power load forecasting method that combines the Variable Mode Decomposition (VMD) algorithm, Convolutional Neural Networks (CNNs), Gated Recurrent Units (GRUs), and an attention mechanism. It is designed to address the limitations of single input features and long-term dependencies in power load forecasting. The attention mechanism effectively identifies key information, providing a more accurate and efficient power load forecasting solution that overcomes several shortcomings of traditional algorithms.

The contributions of this paper are:

1. Traditional power load forecasting methods do not consider the influence of weather factors, such as holidays, seasonal changes, weather, etc., which leads to the accuracy of prediction results being difficult to guarantee. The attention mechanism is introduced to combine power data with meteorological data, and the attention mechanism is used to assign different weights to power features to improve the prediction accuracy.

2. The method proposed in this paper not only uses VMD, CNN, and attention mechanism but also introduces a Gated Recurrent Unit (GRU) for time series modeling. The innovation of this combined model is that it integrates multiple deep learning and signal processing techniques to fully use different levels of data information. The practical application significance is that this integrated model can analyze and model power load data more comprehensively, and improve the ability to capture time correlation, seasonal variation, and nonlinear patterns, thus improving the practical feasibility and accuracy of power load forecasting.

## Related work

After the preliminary analysis of the characteristics of the power load series, Yang X et al., [8] based on the time series method and combined with a support vector machine, built a load forecasting model and analyzed the actual load data of the power grid. The experimental results show that the model has strong practicability. Ran Shen et al. [9] proposed a weather-sensitive load forecasting model to comprehensively evaluate the impact of weather conditions given the huge computational burden of accuracy verification of many algorithms. The load index is analyzed in detail, and the relationship between weather conditions and the load index is evaluated by factor analysis. Then, multivariate nonlinear regression analysis is used to represent the forecasting equation. Qi et al. [10] proposed a short-term power load forecasting method based on an improved exponential smoothing grey model. Firstly, the grey relational analysis method is used to determine the main factors affecting the power load. Then the improved multivariate grey model is used to forecast the power load. Firstly, the original power load data is smoothed by the first exponential smoothing method. Secondly, a grey forecasting model with optimized background value is established by using the smooth sequence that conforms to the exponential trend. Finally, the inverse exponential smoothing method was used to recover the predicted values. The fitting effect of the traditional method is acceptable for the progressive load data, but the prediction effect is poor for the nonlinear load data with large volatility.

Compared with traditional prediction methods, artificial intelligence methods such as gated recurrent units (GRU) and long short-term memory networks (LSTM) have greatly improved the accuracy of load prediction, and the error is reduced by the weight ratio of elements. Compared with traditional algorithms and common machine learning algorithms, these methods have stronger model generalization ability. Shi et al. [11] applied a new deep and improved recurrent neural network based on pooling to family load forecasting and solved the problems of high volatility and uncertainty in traditional methods of family load forecasting by increasing data diversity and data volume. Considering the highly dynamic and stochastic characteristics of a single user's electricity consumption behavior, Hong et al. [12] proposed a learning method based on a deep neural network to analyze the correlation between different electricity consumption behaviors. Rongrong Cai et al. [13] proposed a new short-term load forecasting method for microgrids. After comparing and analyzing all load characteristics in

the time and frequency domains, the training set and text set are selected considering the influence of temperature and day type. Finally, the BP neural network is used to predict the microgrid load. The final results show that the prediction accuracy of the proposed method is significantly better than that of the traditional method. However, directly using artificial intelligence methods to analyze the time series and nonlinearity of load and learn the change law of load, the robustness and accuracy will not be guaranteed because of the noise in data and the model is easy to fall into local optimum.

The prediction accuracy of the artificial intelligence model trained directly on the load data needs to be improved, so the combination forecasting method is applied to load forecasting. The combination of suitable single models can overcome the defects of different models and make full use of their advantages, learning from each other's strengths, while the combination of inappropriate models is likely to amplify their shortcomings, and the accuracy is lower than the single model. Some power load forecasting literature starts from the data itself to reduce the data noise and combine multiple models to improve the prediction accuracy and stability of the model. Wang Delu et al. [14] proposed the fusion of text data and traditional time series data to improve the power demand forecasting ability. Based on the idea of multi-modal information fusion, a new comprehensive power demand forecasting model CNN-LSTM (Convolutional neural Network, long short-term memory) in a multi-heterogeneous data environment was constructed. The empirical results show that the proposed prediction model is effective, which proves that the organic fusion of time series data and text data can effectively improve the prediction performance. However, when processing the fusion of text data and time series data, the algorithm may face the problem of data imbalance. If the number of two data types is different, the model may favor more data types, resulting in unstable performance. Zhuang Zhiyuan et al. [15] artificially reduced the Short-Term Load forecasting (STLF) error of the offline forecasting model, and proposed a VMD-IWOA-LSTM (VIL) method for STLF. Firstly, Variational Mode Decomposition (VMD) was used to decompose historical power load signals. Then, the decomposed signals were reconstructed according to the similarity of the Pearson correlation coefficient (PCC), and the meteorological data were selected for each reconstructed component based on the PCC threshold set. The Long Short-Term Memory (LSTM) model was used to predict the corresponding components, and the improved Whale Optimization Algorithm (IWOA) was used to optimize the parameters in the LSTM. Finally, the prediction results of each component were added to obtain the final prediction result. The experimental results of power load data in a certain area show that the proposed method has the advantages of strong anti-interference performance and high prediction accuracy compared with other methods, and it has strong practicability. However, this single similarity measure may not capture the complex relationship between signals, resulting in other important influencing factors that may be ignored when selecting meteorological data, which affects the final load forecasting accuracy. Tang X et al. [16] proposed a hybrid neural network forecasting model based on deep belief network (DBN) and bidirectional Recurrent neural network (Bi-RNN). The K-means algorithm is used to cluster the load data, and similar data are classified into the same cluster. Then the load data are decomposed into multiple Intrinsic Mode Functions (IMFs) by ensemble Empirical Mode Decomposition (EEMD). Then, the candidate features are selected by calculating the Pearson correlation coefficient, and finally, the prediction input is constructed. Although this method improves the prediction effect, it does not fully consider the weight between features, and the feature extraction is not sufficient. Different from the existing models, we propose the VCAG model in this paper, which analyzes and models the power load data more comprehensively, and improves the ability to capture time correlation, seasonal variation, and nonlinear patterns, thereby improving the practical feasibility and accuracy of power load forecasting.

## Related algorithms

### Parameter optimization

The process of using the particle Swarm optimization (PSO) algorithm to find the optimal parameters of VMD (Variational Mode Decomposition) can be summarized as follows [17]. First, we need to define an objective function, usually an error measure to be minimized, such as mean square error (MSE), to measure the quality of the VMD decomposition. Then, we define the parameter space, determine the search range and constraints for the parameters, and ensure that the parameters vary within a reasonable range. Next, we initialize a set of particles, each representing a parameter vector, and randomly assign a velocity vector. Meanwhile, the individual best position and the global best position of each particle are recorded.

In each iteration of the PSO algorithm, we use a specific formula to update the velocity and position of each particle, which includes inertia weight, acceleration factor, and random number. Then, the current position of each particle and the individual best position are compared, and if the current position is better, the individual best position is updated. Then, the individual best positions of all particles are compared to update the global best position. This process is repeated until the termination condition is met, such as the maximum number of iterations is reached or the error is small enough.

Finally, the parameter vector corresponding to the global best position is considered as the optimal parameter of VMD, and the best decomposition result under the objective function can be obtained by using these parameters to decompose VMD.

### Feature selection

**VMD.**    The regional power load contains periodic, aperiodic, periodic, and trend information. The hidden information in the power load series is difficult to peel away with the naked eye, resulting in a low interpretability of the data. Variational mode decomposition (VMD) is a signal decomposition estimation method. The main idea is to decompose the original signal into several smooth IMFs with different frequencies. VMD determines the frequency center and bandwidth of each IMF by iteratively finding the optimal solution of the variational model, to realize the frequency domain dissection [18] of the signal and IMF adaptively.

VMD can efficiently deal with non-stationary and nonlinear signals such as electric power. By searching and constructing a variational model, the original electric power signal is decomposed into different components with finite width, corresponding to the center frequency of is, and the optimal solution of the variational model is found by alternating iterative updates. $s(t)u_k(t)w_k$ The algorithm is as follows:

In the first step, a Hilbert transform is applied to an electric power component to obtain the one-sided spectrum as $u_k(t)$

$$\left[\delta(t) + \frac{j}{\pi t}\right] \circ u_k(t) \tag{1}$$

n $\delta formula$ ula (3), is Dirac distribution function; ○ represents the convolution operation.

In the second step, the spectrum of the electric power component is transferred to the baseband at the estimated center frequency $e^{-jw_k t}$

$$\left\{[\delta(t) + \frac{j}{\pi t}] \circ u_k(t)\right\} e^{-jw_k t} \tag{2}$$

In the third step, the signal is demodulated by Gaussian smoothing to obtain the broadband of the electrical power component. The variational constraint model can be expressed

as

$$\begin{cases} \sum_{k=1}^{k} \parallel \partial_t\{[\delta(t) + \frac{j}{\pi t}] \circ u_k(t)\}e^{-jw_k t} \parallel_2^2 \\ \sum_{k=1}^{k} u_k = s(t) \end{cases} \tag{3}$$

In Eq (3), $\partial$ (t) means to find the partial derivative; k is the number of components.

In the fourth step, to obtain the optimal solution of the variational problem in Eq (3), the Lagrangian operator $\lambda$ with quadratic multiplicative factor $\alpha$ is introduced. The above constrained variational problem is transformed into an unconstrained variational problem as follows.

$$L(\{u_k\}, \{w_k\}, \lambda) = \alpha \sum_{k=1}^{k} \parallel \partial_t\left\{[\delta(t) + \frac{j}{\pi t}] \circ u_k(t)\right\}e^{-jw_k t} \parallel_2^2 + \parallel s(t) - \sum_{k=1}^{k} u_k(t)$$

$$\parallel_2^2 + \langle \lambda(t), s(t) - \sum_{k=1}^{k} u_k(t) \rangle \tag{4}$$

In the $u_k$ fifth step, each component and its corresponding center frequency are optimized by alternating direction method of multipliers (ADMM). $w_k$

$$\begin{cases} \hat{u}_k^{(n+1)}(\omega) = \dfrac{\hat{s}(\omega) - \sum_{i \neq k} \hat{u}_k^n(\omega) + \dfrac{\hat{\lambda}^{(n)}(\omega)}{2}}{1 + 2\alpha(\omega - \omega_k^{(n)})^2} \\ \\ \omega_k^{(n+1)} = \dfrac{\int\limits_{0}^{\infty} \omega |\hat{u}_k^{(n+1)}(\omega)|^2 d\omega}{\int\limits_{0}^{\infty} |\hat{u}_k^{(n+1)}(\omega)|^2 d\omega} \end{cases} \tag{5}$$

In $\hat{s}(\omega)$ Eq (7), n is the number of iterations;,,, stands for the Fourier transform of s(t),, respectively. $\hat{u}(\omega)\hat{\lambda}(\omega)u_k(t)\lambda(t)$

**CNN.** Historical PV data contain few temporal information features at a single time scale, which cannot fully reflect the information and trends of the time series. It is necessary to obtain more time series features from raw power data and meteorological elements. Convolutional Neural Networks (CNN) have excellent feature extraction ability and are widely used in speech-emotion recognition and face recognition.

In recent years, researchers have improved feature extraction [19,20] by applying convolutional neural networks to time series data. The essence is to use filters to extract features from the data to obtain feature vectors and use activation functions to solve classification or regression problems [21]. In this paper, 3D convolution is used to extract features from raw time series data with the following formula:

$$y_t = \sum_{k=1}^{K} w_t x_{t-k+1} + b \tag{6}$$

Where is the output feature data; $y_t w_t$ Is the convolution kernel; $x_{t-k+1}$ Is the input data; b is for bias; k is the data length.

## Model training

**Attention mechanism.** The attention mechanism can help the model pay more attention to certain variables or features during the analysis process, to better predict future weather

conditions. When dealing with meteorological data, the attention mechanism can be used to focus the model's attention on certain important meteorological features, such as temperature, humidity, and air pressure, to predict future weather conditions more accurately.

Although GRU can describe the internal relationship of electric power output data, it is affected by extreme weather changes, which leads to large fluctuations in electric power output. In the case of large electric power fluctuations, the accuracy of prediction using a single GRU neural network is often not high. The attention mechanism imitates the attention model of the human brain and focuses attention on specific areas. Therefore, this paper can use the attention mechanism to give different contribution rates to the input power characteristics to improve the prediction accuracy.

**GRU.** By training a GRU model and using historical power load data, we can learn the dynamic change pattern of power load data and use the model to predict the change of power load in future time steps. Gated Recurrent Unit (GRU), a variant of recurrent neural network (RNN), is a powerful sequential modeling tool that can be used for time series modeling and forecasting. It can capture long-term dependencies in time series data. It is used to control the flow and retention of information.

In power load forecasting, we use the GRU model to capture the change in power load data. $L = \{L_1, L_2, \ldots, L_t\}$ Power load data usually changes over time, and we can represent power load data as a time series, denoted as, where t stands for time step. GRU model can be used to model the change process of this time series.

The basic structure of GRU includes an update gate and a reset gate, which function as follows:

Update gate: Controls how much information from the previous moment is retained at the current moment. Its output value is a number between 0 and 1, which is used to weigh the state at the previous time and the input at the current time. When the output value of the update gate is close to 1, the previous state is completely preserved; When the output value is close to 0, the state at the previous moment is completely ignored. The update gate is calculated as follows.

$$z_t = \sigma(W(z) * [h_{t-1}, x_t]) \tag{7}$$

Where is the input at the current time, is the state at the previous time, is the learnable weight matrix, and is the sigmoid function? $x_t h_{t-1} W(z) \sigma$

Reset gate: controls how much information from the previous moment is forgotten at the current moment. Its output value is again a number between 0 and 1, which is used to weigh the previous state and the current input. When the output value of the reset gate is close to 1, the previous state is completely preserved; When the output value is close to 0, the previous state is completely forgotten. The reset gate is calculated as follows.

$$r_t = \sigma(W_r * [h_{t-1}, x_t]) \tag{8}$$

Where is the learnable weight matrix. $W_r$

Based on the update gate and reset gate, we can calculate the candidate state at the current moment as follows $\hat{h}_t$

$$\hat{h}_t = \tanh(W_h * [r_t, \odot h_{t-1}, X_t]) \tag{9}$$

Where $\odot$ represents element-wise multiplication and is a learnable weight matrix. $W_h$

Finally, we can calculate the state at the current instant by updating the gate and candidate states as follows: $h_t$

$$h_t = (1 - Z_t) \odot h_{t-1} + Z_t \odot \hat{h}_t \tag{10}$$

Here, $\odot$ again means element-wise multiplication. The meaning of this formula is that if the output value of the update gate is close to 1, then the state at the previous moment is completely preserved; $h_{t-1}$ If the output value of the update gate is close to 0, then the candidate state at the current time is completely preserved. $\hat{h}_t$ In this way, GRU can effectively control the flow of information, avoid the long-term dependence problem, and have fewer parameters and faster training speed.

These formulas describe how the GRU model updates the hidden state at the current time step based on the past hidden states and the current input. $h_{t-1} x_t h_t$ In power load forecasting, the hidden state can be viewed as a characteristic representation of the power load at the current time step, and the reset gate and update gate determine how to fuse the past and current information for forecasting. $h_t$

## Power load forecasting methods

Traditional methods are difficult to deal with nonlinear and non-stationary time series data and have defects in extracting key features, recognizing complex patterns, and remembering and using long-term time dependencies. The algorithm structure proposed in this paper can solve the limitations of traditional algorithms in data complexity, processing power, and prediction accuracy by integrating multiple advanced technologies. Firstly, it is necessary to collect historical power load data and preprocess them. Preprocessing includes data cleaning, denoising, normalization, and other operations so that the model can better learn and predict. Next, the Variable Mode decomposition (VMD) technique was used to decompose the power load data. The VMD technique can decompose the signal into multiple fixed-width bandpass band components, which can describe the different frequency components of the signal. This step can improve the accuracy of the model so that it can better capture the time and frequency characteristics of power load data. A Convolutional Neural Network (CNN) is used to extract the features of the decomposed signals. CNN can automatically learn the spatio-temporal features in power load data, identify the patterns of different frequency components, and extract their relevant features. To improve the accuracy and robustness of the model, the Attention mechanism (AM) is used to weigh the features extracted by CNN. AM can dynamically adjust the weight of CNN features according to the importance of the input data, to better capture the important features of the data.

Then, the Gated Recurrent Unit (GRU) was used to model the dynamics and long-term dependence of time series data, and the hidden state size of GRU was 100, and the bidirectional GRU was used for modeling. A fully connected layer was added to the top of the GRU as the output layer to map the hidden state of the GRU unit to the predicted value. Finally, through training and optimization, the Adam optimizer was used to update the parameters, and the Mean Square Error (MSE) was used as the loss function for optimization. The specific model is shown in the Fig 1 below.

## Experimental design and analysis

In this section, we first introduce the dataset of the experiment and how to preprocess the dataset and the weather factors. Secondly, the vmd-cnn-am-GRU model proposed in this paper and the experimental process are introduced.

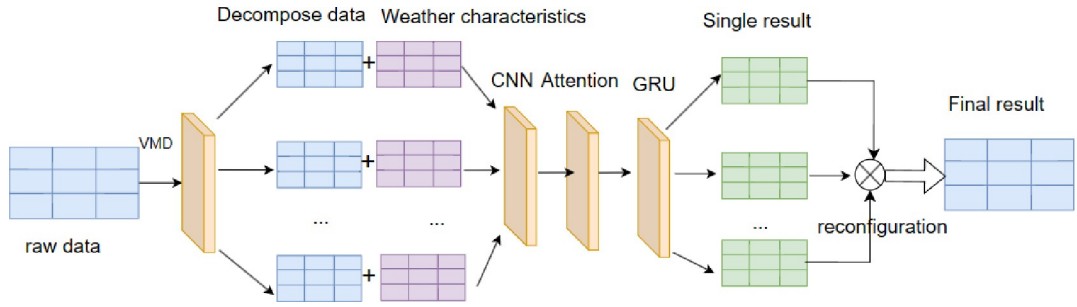

**Fig 1. VCAG model.**

## Dataset

In this paper, two data sets are used for experimental testing. The first data set is the power load forecasting sample data, which contains the actual load data on December 31, 2018, and December 22, 2019, and the load value is recorded every hour. It also provides daily weather data for the two years. A variety of meteorological factors include wind speed, wind direction, pressure, and temperature. Another load data is also the network open source power data. The load data is the electricity consumption of a place from September 2, 2013, to August 31, 2015, and the load value is recorded every hour. It includes factors such as daily weather conditions and temperature. The details are shown in Table 1. Specific reference can be made https://github.com/yzc-dot/Electricity-prediction.git.

## Data processing

Due to the volatility of real electricity data, a large amount of data can cause numerical problems. In the first data set, the load value is recorded every hour, and there is a phenomenon of missing data within one hour. In this case, the mean imputation method is used to fill in the missing data. Using the VMD algorithm, the non-flat data is transformed into multiple stationary data. Standardization of power load data: We normalize the power data to [0,1], and the calculation process is as follows:

$$X_{norm} = \frac{x - x_{min}}{x_{max} - x_{min}} \tag{11}$$

Where and are the minimum and maximum values in the power data, respectively? $x_{min}$ $x_{max}$ X is the value of the unprocessed power load and is the normalized value. $X_{norm}$ The training set and validation set are used to train the model, the data set is divided, and the test set is input into the trained model for prediction.

## Evaluation metrics

We learned that in the power system planning stage, if the load forecast result is too low, the system installed capacity is insufficient to meet the residential electricity demand, and even the power shortage will lead to a power outage. If the load forecast result is too high, it will lead to

**Table 1. Dataset.**

| Datasets | sampling interval | characteristic | Data volume per column | Dimension |
|---|---|---|---|---|
| 1 | 1h | 5 | 8568 | 5 * 8568 |
| 2 | 1h | 4 | 17496 | 4 * 17496 |

inefficient operation of power generation and transmission equipment, resulting in a waste of investment. To make our experimental results more convincing, evaluation indicators such as root Mean Square error (RMSE), Mean absolute error (MAE), and coefficient of determination (R2) are used, where smaller values of RMSE and MAE indicate better prediction performance of the model. The range of R2 is 0 to 1. The closer a value is to 1, the more explanatory the variables are for y and the better fit the model is for the data. The Eqs (12)–(14) are as follows:

$$RMSE = \sqrt{\frac{1}{n}\sum_{i=1}^{n}(y_i - y_{ik})^2} \tag{12}$$

$$MAE = \frac{1}{n}\sum_{i=1}^{n}|y_i - y_{ik}| \tag{13}$$

$$R^2 = 1 - \frac{\sum_{i=0}^{M}(y_i - \hat{y}_i)^2}{\sum_{i=0}^{m}(y_i - \bar{y}_t)^2} \tag{14}$$

Where is the ith predicted value of the model, is the ith actual value, y is the mean of the test sample, $y_i y_{ik} \hat{y}_t$ is the estimated value $\bar{y}_t$, is the mean, and n is the number of test samples.

## Experimental procedure

**Electric power decomposition results based on VMD.** The power load values for the two datasets are shown in Figs 2 and 3.

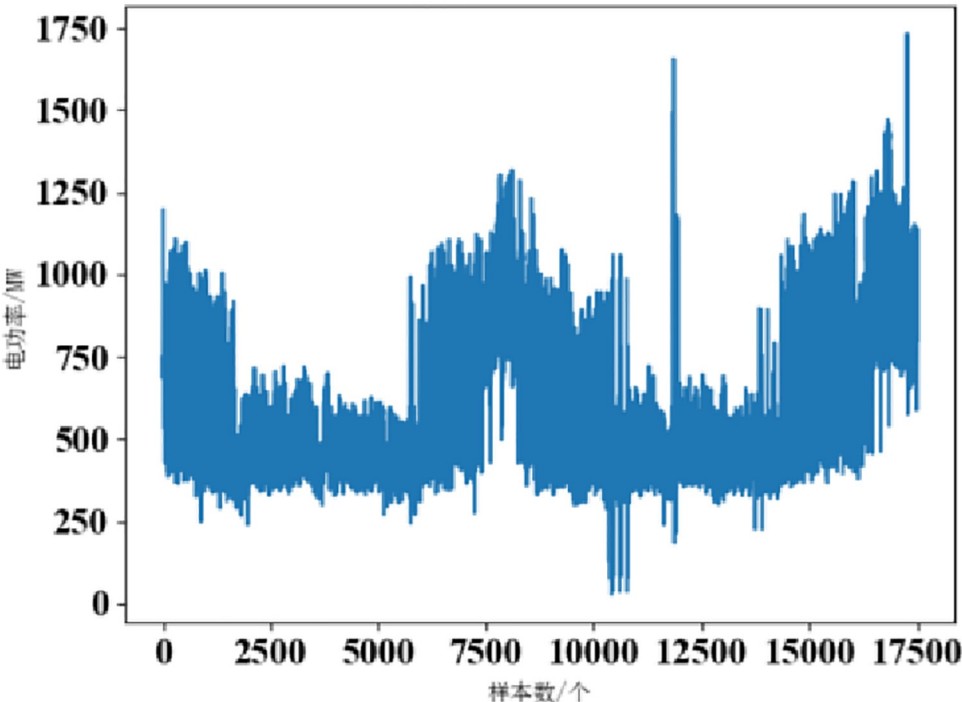

原始电功率趋势图

**Fig 2. Dataset 1.**

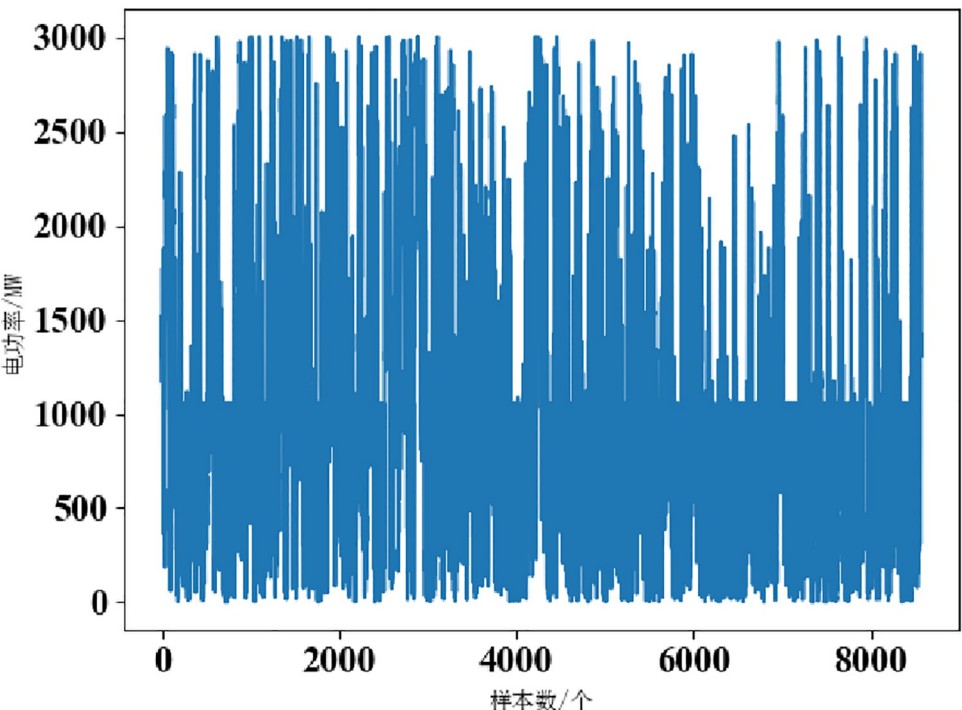

原始电功率趋势图

**Fig 3. Dataset 2.**

By looking at dataset 1, it is clear that the data fluctuates drastically and has a strong randomness. The maximum value in the data is 1700 MW, the minimum value is 500 MW, the average value and standard deviation of 17500 data are 789.2709852024923 and 223.94484108659873, respectively, the difference is about 566MW, and the degree of dispersion is proportional to the difference. Therefore, the electric power has strong randomness and volatility. Using VMD to decompose the data can make good use of its multi-frequency characteristics.

The electric power sequence is decomposed into different components by VMD, and the corresponding k center frequencies are obtained by setting k different components. Mainly for the selection of parameters k and values, the particle swarm optimization algorithm is used to optimize the vmd algorithm, and finally, the values of k and a are determined to be 4 and 2828.18 respectively. For the power dataset, the final values of k and a are 3 and 992.17, respectively. The specific decomposition results are shown in Figs 4 and 5.

**Training process.** Firstly, a new data set is composed of the above data and meteorological factors after vmd decomposition, of which 80% is used as the training set and 20% is used as the test set. Through normalization, the value range of different features in the training set can be unified to between 0–1, which makes the distribution of data more uniform and more in line with the requirements of the algorithm, and improves the generalization ability and accuracy of the model. The normalized data were passed into the model, and the model required the input to be a three-dimensional tensor with the shape of (60,4). The data of (17500,4) were converted into a three-dimensional array, in which the shape of each two-dimensional array was (60,4). The input is processed through the GRU layer and the hidden state at each time step is returned. After that, the output of the GRU layer was convolved, and

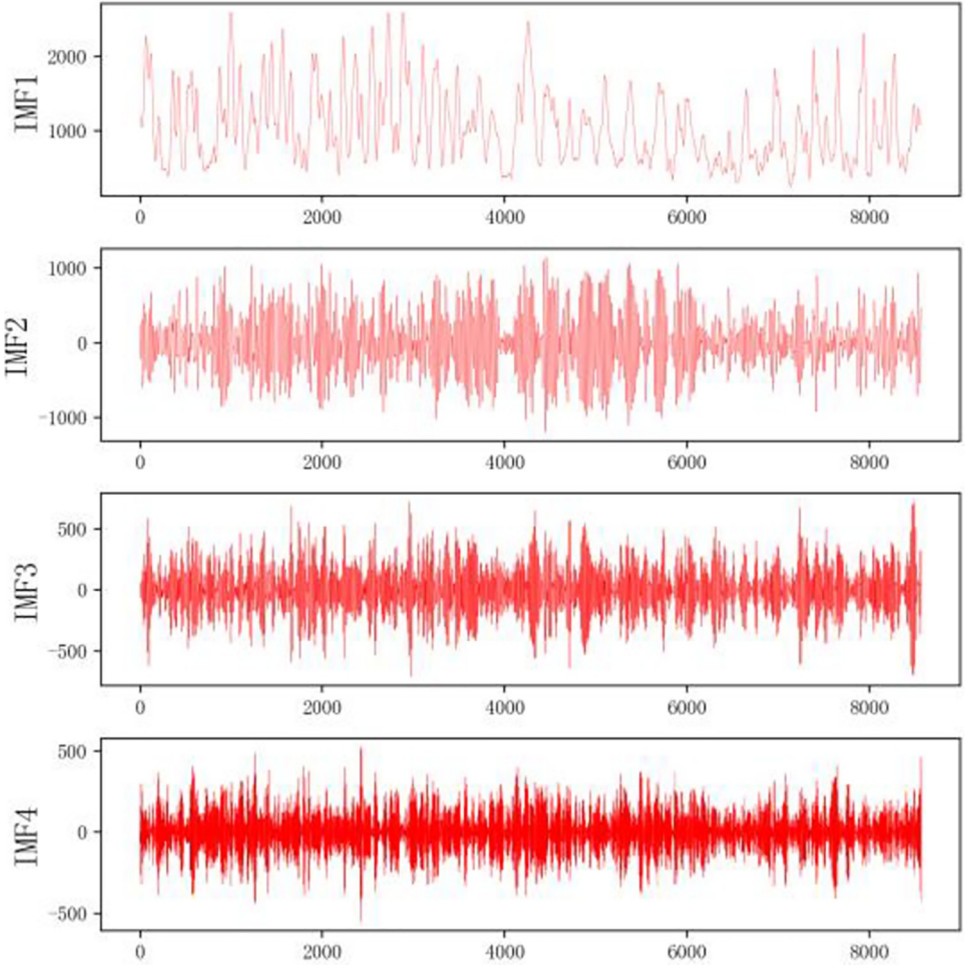

**Fig 4. VMD breakdown diagram of the dataset 1.**

the one-dimensional convolution was used for feature extraction. The convolution kernel size was 5*5, and the step size was 1. The one-dimensional convolution was used again to extract the features of the previous step, and the size of the convolution kernel was 3*3. Then, the extracted features were input into the pooling layer, and the Max pooling operation was performed on the convolution results. Then, the output results of the pooling layer are processed by the attention mechanism to improve the model's attention to key information. After the attention layer, a bidirectional GRU layer is defined to further extract features, where the units are set to 100, indicating that the number of hidden layer units is 100. Because it is a bidirectional GRU, the output dimension is 2*100 = 200. Define a bidirectional GRU layer again, the output shape of this layer is (1, 200), the output is the concatenation of the hidden states of the last time step, where 200 is the number of LSTM units, 200 represents the dimension of the hidden state, because it is a bidirectional GRU, so the output dimension is 2*200 = 400. The bidirectional GRU can consider the past and future information at the same time, which is beneficial to capture the long-term dependence of time series data. After that, the data is flattened to change the shape from (1, 400) to (400,). All the information from the previous step is extracted, which can be further used in the training and prediction of the neural network. After that, we define a fully connected layer whose output shape is (25,). Define another output

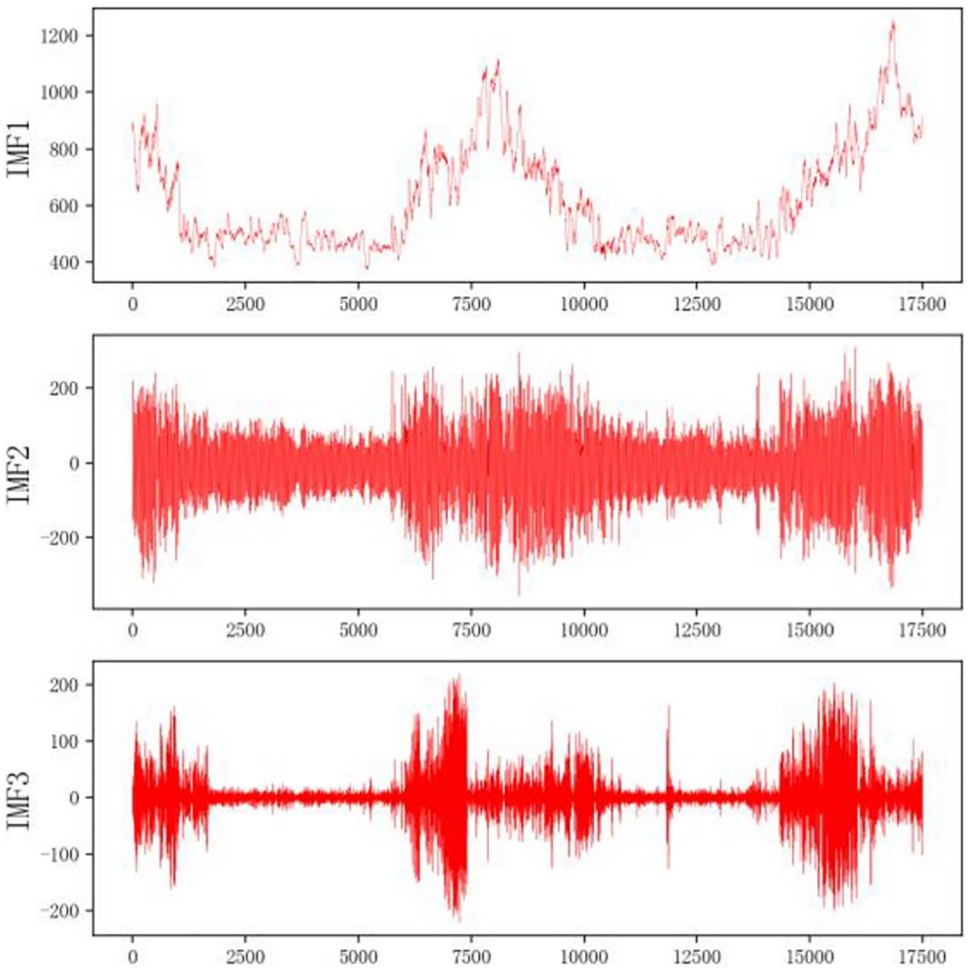

**Fig 5. VMD Breakdown diagram of dataset 2.**

layer with an output shape of (1,). The input goes through a hidden layer of 25 neurons, and then through an output layer of 1 neuron to output the prediction result. The specific parameters are shown in Table 2:

We first use the mean squared error loss when training the model, which is mathematically expressed as $L(y, \hat{y}) = \Sigma(y_i - \hat{y}_i)^2$ where y is the true label and $\hat{y}$ is the predicted value of the model. The final total error is the sum of the loss over each training run. This loss function is used to measure the gap between the predicted value of the model and the true label. The

**Table 2. Model parameters.**

| parameters | 1 | 2 |
|---|---|---|
| Epoch | 100 | 100 |
| timesteps | 60 | 60 |
| batch_size | 32 | 32 |
| Convolutional kernel size | 5\3 | 5\3 |
| Number of convolution kernels | 5 | 4 |
| Initial learning rate | 0.0001 | 0.0001 |
| Dropout | 0.01 | 0.01 |

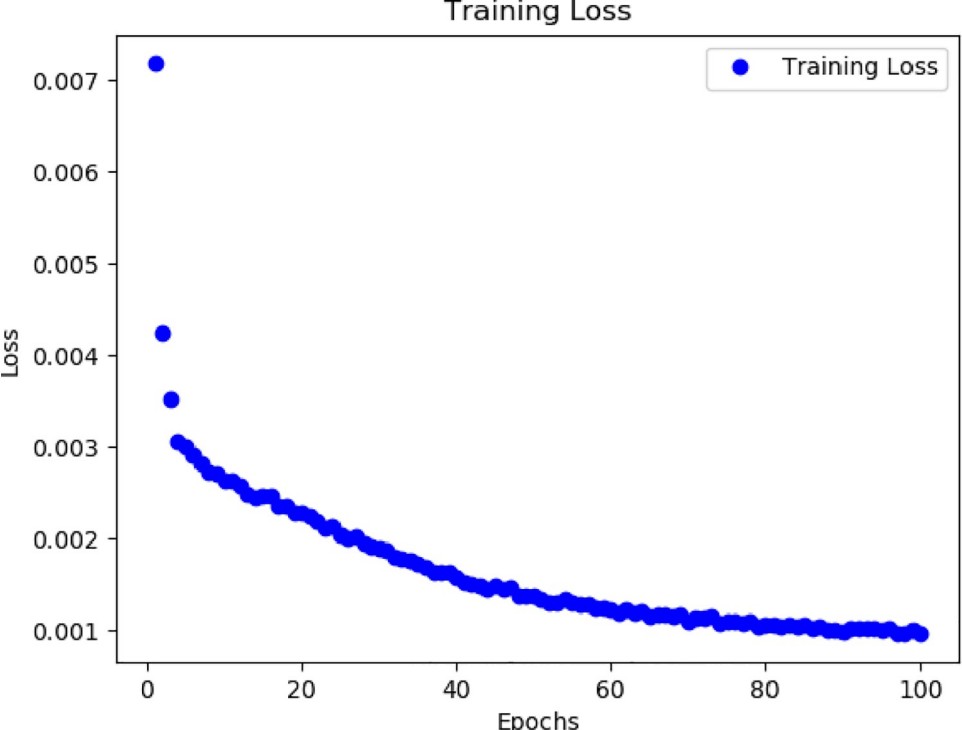

**Fig 6. Dataset 1 training loss.**

figure below shows the loss of the model on the training set as a function of the number of training iterations. As you can see from Figs 6 and 7, the loss decreases as the training progresses, indicating that the model gradually learns to fit the training data better.

We observe that lower loss values are generally associated with better model performance, but not absolutely. Sometimes, even with low loss values, the model may still perform poorly on certain tasks, possibly due to overfitting or other factors. Therefore, we do not only rely on the loss value when evaluating the model but also consider other performance metrics to fully assess the performance of the model.

## Prediction Results and comparative analysis based on VCAG

To better prove the prediction ability and generalization ability of the VCAG model proposed in this paper, a single BP, GRU, Bert, and LSTM model, as well as the combined models CNN-GRU and CNN-AM-GRU are established to predict the power load value. The prediction results are compared with the prediction results of the model proposed in this paper. From Figs 8–10, it can be seen that the trend of the electricity power prediction curve of the proposed VCAG model is consistent with the actual power curve, mainly because our model analyzes and models the power load data more comprehensively, improves the ability to capture time correlation, seasonal changes, and nonlinear patterns, and thus improves the practical feasibility and accuracy of power load forecasting.

To better reflect the superiority of the VMD-CNN-AM-GRU prediction model proposed in this paper, we compare the errors between the predicted value of electricity consumption and the actual value of the single BP, GRU, Bert, and LSTM models, as well as the combined model CNN-GRU and CNN-AM-GRU prediction models, and the comparison results are shown in Table 3. It can be seen from the table that compared with the single BP, Bert, and GRU electric

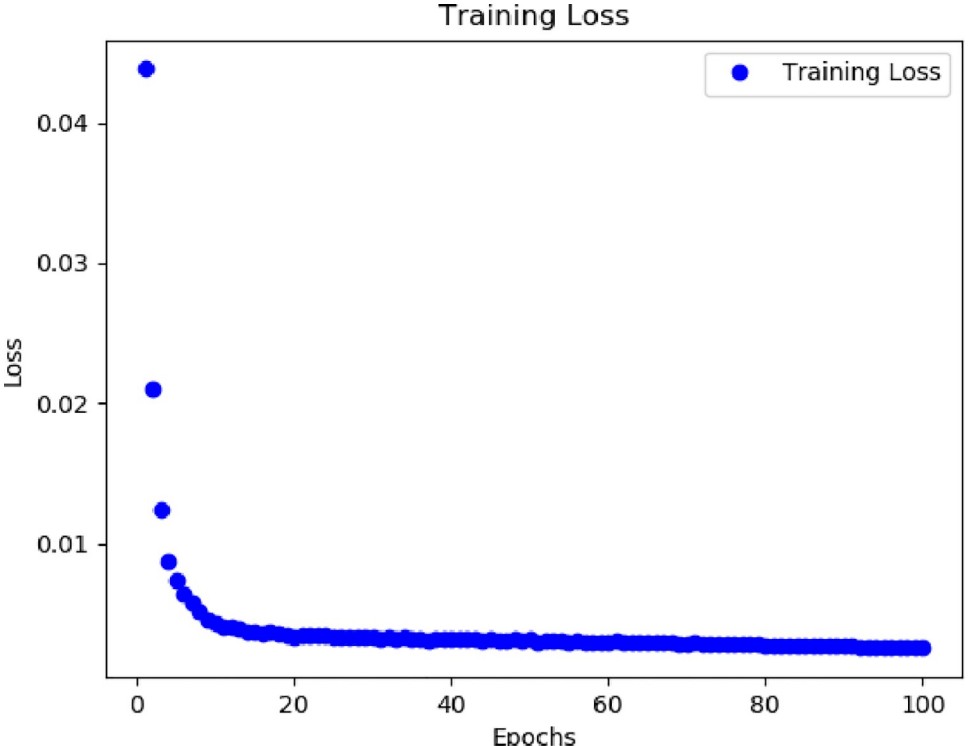

**Fig 7. Dataset 2 training loss.**

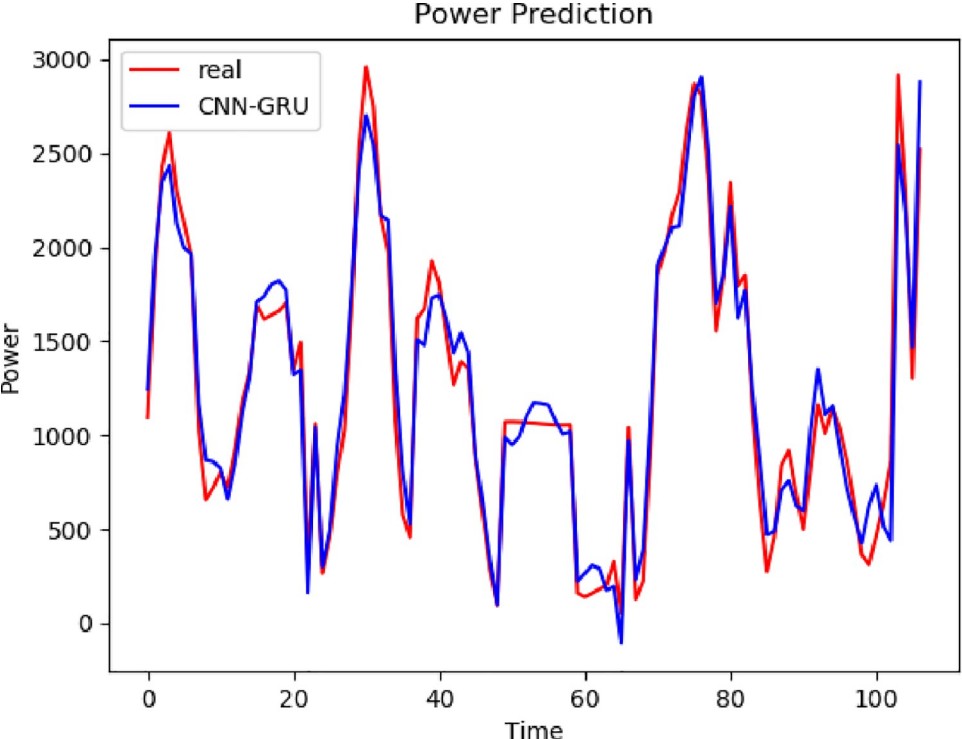

**Fig 8. Results of VMD-CNN-AM-GRU.**

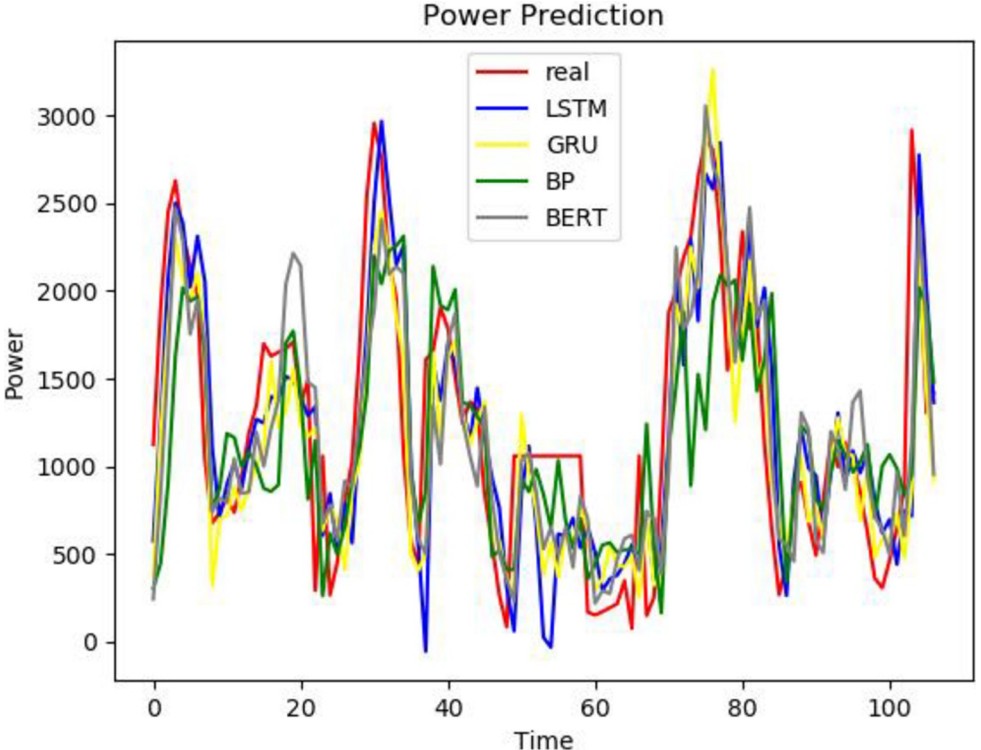

**Fig 9. Single model.**

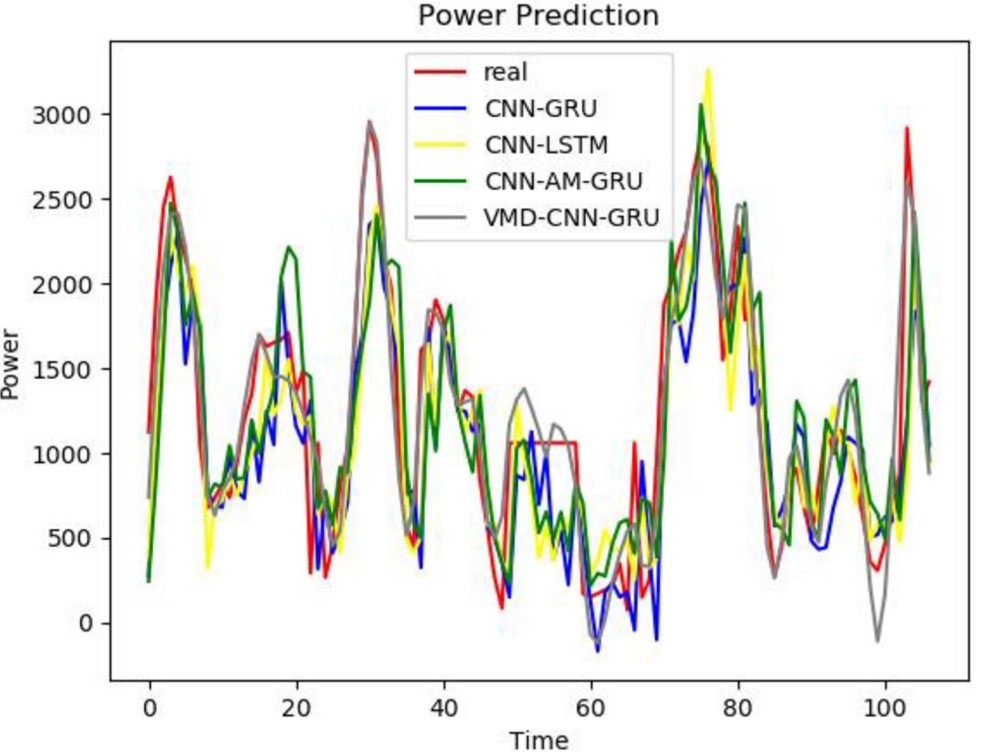

**Fig 10. Combined models.**

**Table 3. Data set 1 evaluation index results.**

|  | MAE | RMSE | R2 |
|---|---|---|---|
| LSTM | 378.1624551548411 | 517.5815586913018 | 0.5301492256236884 |
| GRU | 376.91025616856086 | 511.6528832541347 | 0.5602494957052369 |
| BERT [22] | 377.91025616856086 | 511.7281343406944 | 0.5476647961235788 |
| BP [23] | 378.5568236159252 | 512.3645464911516 | 0.5268559917866708 |
| CNN-LSTM [24] | 358.5568236159252 | 485.351439285301 | 0.5859366394148568 |
| CNN-GRU | 356.253519783215 | 477.5981229395852 | 0.5999377425578861 |
| VMD-CNN-GRU | 55.288781531261684 | 75.30088441627616 | 0.9906878452301766 |
| VMD-CNN-AM-GRU | 20.280231174158864 | 23.881207789211846 | 0.9989997353207647 |

power prediction models, the prediction accuracy of the three groups of models corresponding to the VMD decomposition algorithm is relatively improved, which proves that the combined model is superior to the single prediction model. Compared with CNN-GRU, the MAE of VMD-CNN-GRU is reduced by 303.27 and RMSE is reduced by 402.29MW, which proves that the prediction accuracy is improved by adding the VMD algorithm. Because the MAE of VCAG of the model in this paper is reduced by 34.97 and RMSE is 51.42MW compared with VMD-CNN-GRU, it is proved that the prediction model with attention mechanism reduces the prediction error. It can be seen that in the problem of electric power prediction, the hybrid prediction model of variational mode decomposition, attention mechanism, and long short-term memory neural network has better prediction performance.

The same data set 2 is predicted using different models, and the prediction results of different models are compared with the prediction results of the model proposed in this paper. It can be seen from Figs 11–13 that the trend of the electricity power prediction curve of the

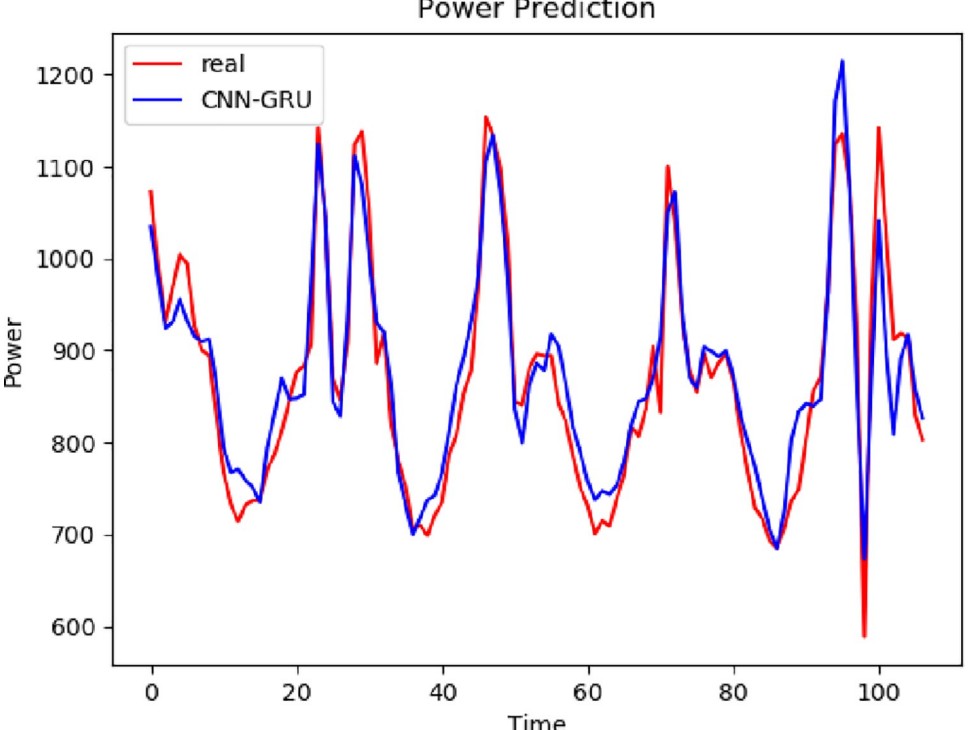

**Fig 11. VCAG.**

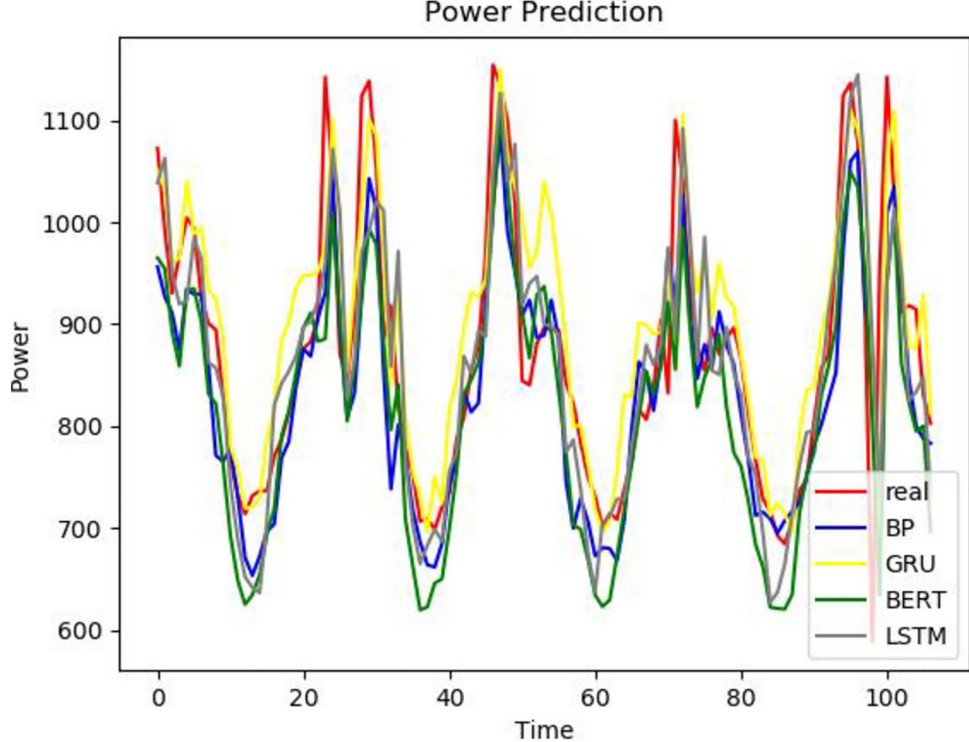

**Fig 12. Single model.**

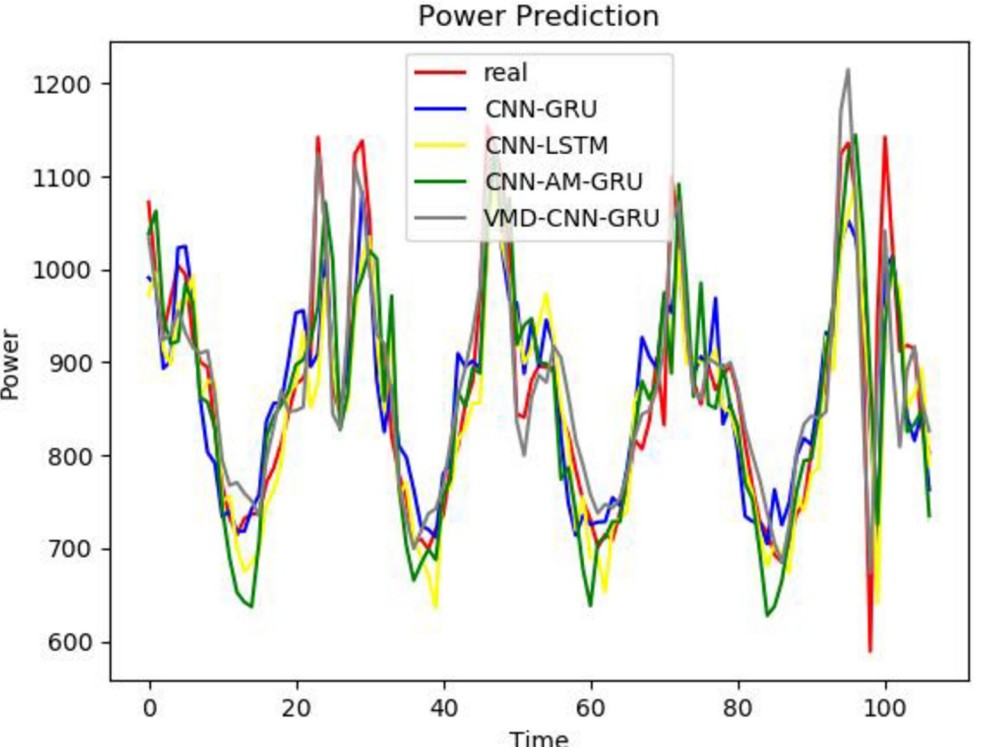

**Fig 13. Combined model.**

**Table 4. Data set 2 evaluation index results.**

|  | MAE | RMSE | R2 |
|---|---|---|---|
| LSTM | 66.90731364489719 | 86.26923750627041 | 0.5411458782684222 |
| GRU | 56.66997869155141 | 76.1676953359481 | 0.6563376596261866 |
| BERT [22] | 52.31486598143588 | 72.1107546789745 | 0.6966775446689624 |
| BP [23] | 53.50414850471025 | 74.9748469241702 | 0.6534276640279069 |
| CNN-LSTM [24] | 52.22276598143925 | 71.0170796348745 | 0.6890515927027088 |
| CNN-GRU | 51.65861588786915 | 74.65942015095665 | 0.6423120182754858 |
| VMD-CNN-GRU | 11.602667400729006 | 14.06421909802147 | 0.9878046585346968 |
| VMD-CNN-AM-GRU | 6.0957062480374 | 8.340781585166098 | 0.9957108008069071 |

proposed VCAG model is consistent with the actual power curve. It proves once again that the accuracy of the proposed model is high.

In the second data set, each model evaluation index is shown in the Table 4. It can be seen from the table that compared with the single CNN-GRU and GRU electric power prediction models, the prediction accuracy of the three groups of models corresponding to the VMD decomposition algorithm is relatively improved, which also proves that the combined model is superior to the single prediction model. Compared with CNN-GRU, the MAE of VMD-CNN-GRU is reduced by 40.05 and RMSE is 62.59MW, which proves that the addition of the VMD algorithm improves the prediction accuracy. Compared with VMD-CNN-GRU, the MAE of VMD-CNN-AM-GRU is reduced by 5.51 and RMSE is 5.72MW, which proves that the prediction model with attention mechanism reduces the prediction error. It can be seen that in the problem of electric power prediction, the hybrid prediction model of variational mode decomposition, attention mechanism, and long short-term memory neural network has better prediction performance.

## Conclusions

This paper aims to solve the problems existing in traditional power load forecasting methods, including the reduced accuracy caused by multiple factors and insufficient feature extraction. To this end, we propose an innovative combined power load forecasting method based on Variable mode decomposition (VMD), convolutional neural Network (CNN), attention mechanism, and Gated Recurrent Unit (GRU), referred to as VCAG. Through multi-level time-frequency feature extraction and time series modeling of power load data, this paper realizes more accurate and stable power load forecasting.

Firstly, we adopt VMD technology to decompose power load data into different time-frequency components, which enables us to better capture various changes and fluctuations in the data. These time-frequency components are used as the input of CNN to further improve the representation ability of the data. Secondly, the attention mechanism is introduced to weight the features output by CNN in a targeted way, so that important features receive higher weights. This helps the model to better focus on the key factors that affect the power load, thereby improving the accuracy of prediction. Finally, we use the GRU network for time series modeling, which combines features with time correlation to further improve the stability and reliability of the forecast.

Through experimental verification on two public data sets, our model shows excellent performance in power load forecasting. The experimental results show that the VCAG method not only has high accuracy but also has excellent stability, which can effectively solve the problems existing in traditional methods. Therefore, the method of this study has a wide range of

application prospects, which can provide more accurate and reliable prediction results for practical applications in the field of power load forecasting. At the same time, it also provides a powerful method and idea for the further study of multi-scale analysis and high-dimensional data processing capabilities.

## Author Contributions

**Funding acquisition:** Haoyue Sun.

**Writing – original draft:** Bining Zhang.

**Writing – review & editing:** Zhicheng Yu.

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
