## [Decision Letter · Decision Letter 0]

24 Nov 2023

PONE-D-23-36467Research on short-term power load forecasting based on VMD and GRUPLOS ONE

Dear Dr. yu,

Thank you for submitting your manuscript to PLOS ONE. After careful consideration, we feel that it has merit but does not fully meet PLOS ONE’s publication criteria as it currently stands. Therefore, we invite you to submit a revised version of the manuscript that addresses the points raised during the review process.

We look forward to receiving your revised manuscript.

Kind regards,

Nasir Ayub, Ph.D.

Academic Editor

PLOS ONE

Journal Requirements:

3. "Funding Information and Financial Disclosure sections do not match:

""We note that the grant information you provided in the ‘Funding Information’ and ‘Financial Disclosure’ sections do not match. 

"none" 

6. Please amend the manuscript submission data (via Edit Submission) to include author yajun liu and xinxin yin.

7. Please ensure that you refer to Figure 1,2,3,4,6,8,9,10,11,12,13,14 and 15 in your text as, if accepted, production will need this reference to link the reader to the figure.

8. We note you have included a table to which you do not refer in the text of your manuscript. Please ensure that you refer to Table 1 and 2 in your text; if accepted, production will need this reference to link the reader to the Table.

Additional Editor Comments:

Dear Author,

I am writing to inform you that we have received the reviewers' comments on your submitted article. The feedback suggests the need for revisions, particularly in incorporating and refining the novelty of the content.

We kindly request your collaboration in addressing the reviewers' comments and making the necessary modifications to enhance the article's overall quality. Once the revisions are complete, our editorial team will carefully evaluate the changes to determine the article's suitability for publication.

Should you have any questions or require further guidance on specific comments, please feel free to reach out.

Reviewers' comments:

Reviewer's Responses to Questions

**Comments to the Author**

1. Is the manuscript technically sound, and do the data support the conclusions?

Reviewer #1: Partly

Reviewer #2: Yes

2. Has the statistical analysis been performed appropriately and rigorously? 

Reviewer #1: No

Reviewer #2: No

3. Have the authors made all data underlying the findings in their manuscript fully available?

Reviewer #1: No

Reviewer #2: Yes

4. Is the manuscript presented in an intelligible fashion and written in standard English?

Reviewer #1: No

Reviewer #2: No

5. Review Comments to the Author

Reviewer #1: The paper is of poor quality and cannot be published in its current form, and the proposed method is not innovative enough.

1. The quality of the picture is poor, please modify it.

2. Grammar issues are an important issue currently.

3. There are many formatting errors in references.

4. The comparison method is not novel enough and it is difficult to demonstrate the effectiveness of the proposed method.

5. Insufficient literature analysis and few references.

Reviewer #2: This article requires major revisions.

1. the author needs to revise the manuscript English and made clearly the contributions, as more research is already carried out in this area.

2. The references are not in a correct format even including their citation.

3. perform statistical analysis, log loss.

4. Compare it with BERT Model.

5. Revise the major sections accordingly.

6. extensive English changes requires.

7. correct the shape of the article. it seems a report rather than a research article.

6. PLOS authors have the option to publish the peer review history of their article (what does this mean?). If published, this will include your full peer review and any attached files.

Reviewer #1: No

Reviewer #2: **Yes: **Ch Anwar ul Hassan

---

## [Author Response · Author response to Decision Letter 0]

13 Feb 2024

Reviewer #1: 

1. Answer： The paper is of poor quality and cannot be published in its current form, and the proposed method is not innovative enough.

Response: Thank the reviewers for their valuable suggestions. We have improved the content of the paper, but our previous writing was not comprehensive enough, and we have added some experiments to prove the innovation of our experiments.

2. Answer： The quality of the picture is poor, please modify it.

Response: Thank the reviewers for their valuable suggestions. We redrew the model drawing to improve the picture quality, and the specific modifications are shown in Figure 1.

Figure 1 VCAG model

3 Answer： Grammar issues are an important issue currently.

Response: Thank the reviewers for their valuable suggestions. We have corrected the grammar errors and also asked my supervisor to help review them.

1. De Giorgi, M.G.; Congedo, P.M.; Malvoni, M. Photovoltaic power forecasting using statistical methods: Impact of weather data.IET Sci. Meas. Technol. 2014, 8, 90–97. 

2. Sansa, I.; Boussaada, Z.; Bellaaj, N.M. Solar Radiation Prediction Using a Novel Hybrid Model of ARMA and NARX. Energies 2021, 14, 6920

3. Jiang, Y.; Zheng, L.; Ding, X. Ultra-short-term prediction of photovoltaic output based on an LSTM-ARMA combined model driven by EEMD. J. Renew. Sustain. Energy 2021, 13, 046103.

4. Answer： There are many formatting errors in references.

Response: Thank the reviewers for their valuable suggestions. We have made modifications to the references and added some additional ones.

5. Answer： The comparison method is not novel enough and it is difficult to demonstrate the effectiveness of the proposed method.

Response: Thank the reviewers for their valuable suggestions. we have modified the comparison method and added several methods for comparison.

 MAE RMSE R2

LSTM 66.90731364489719 86.26923750627041 0.5411458782684222

GRU 56.66997869155141 76.1676953359481 0.6563376596261866

BERT[22] 52.31486598143588 72.1107546789745 0.6966775446689624

BP[23] 53.50414850471025 74.9748469241702 0.6534276640279069

CNN-LSTM[24] 52.22276598143925 71.0170796348745 0.6890515927027088

CNN-GRU 51.65861588786915 74.65942015095665 0.6423120182754858

VMD-CNN-GRU 11.602667400729006 14.06421909802147 0.9878046585346968

VMD-CNN-AM-GRU 6.0957062480374 8.340781585166098 0.9957108008069071

6. Answer： Insufficient literature analysis and few references.

Response: Thank the reviewers for their valuable suggestions. We have added additional references and compared our algorithm with the algorithms in the literature.

Reviewer #2: This article requires major revisions.

1. Answer：the author needs to revise the manuscript English and made clearly the contributions, as more research is already carried out in this area.

Response: Thank the reviewers for their valuable suggestions. We have revised some unclear areas in the initial draft and highlighted the innovation of this article.

2. Answer The references are not in a correct format even including their citation.

Response: Thank the reviewers for their valuable suggestions. There were indeed errors in the previous literature. We have modified the reference format and added some new references.

3. Answer: perform statistical analysis, log loss.

Response: Thank the reviewers for their valuable suggestions. We have added the calculation of losses in the article and provided the losses during the training process.

4. Answer: Compare it with BERT Model.

Response: Thank the reviewers for their valuable suggestions. We used the BERT model for prediction and included the scores in the evaluation table.

 MAE RMSE R2

LSTM 66.90731364489719 86.26923750627041 0.5411458782684222

GRU 56.66997869155141 76.1676953359481 0.6563376596261866

BERT[22] 52.31486598143588 72.1107546789745 0.6966775446689624

BP[23] 53.50414850471025 74.9748469241702 0.6534276640279069

CNN-LSTM[24] 52.22276598143925 71.0170796348745 0.6890515927027088

CNN-GRU 51.65861588786915 74.65942015095665 0.6423120182754858

VMD-CNN-GRU 11.602667400729006 14.06421909802147 0.9878046585346968

VMD-CNN-AM-GRU 6.0957062480374 8.340781585166098 0.9957108008069071

5. Answer: Revise the major sections accordingly.

Response: Thank the reviewers for their valuable suggestions. We have adjusted the content of some chapters and also modified the content of some chapters.

6. Answer: extensive English changes requires.

Response: Thank the reviewers for their valuable suggestions.We have corrected the grammar of English and polished it again

7. Answer: correct the shape of the article. it seems a report rather than a research article.

Response: Thank the reviewers for their valuable suggestions. We have adjusted the chapters and also made some modifications to make the article more like an academic paper

---

## [Decision Letter · Decision Letter 1]

1 Apr 2024

PONE-D-23-36467R1Research on short-term power load forecasting based on VMD and GRUPLOS ONE

Dear Dr. yu,

Thank you for submitting your manuscript to PLOS ONE. After careful consideration, we feel that it has merit but does not fully meet PLOS ONE’s publication criteria as it currently stands. Therefore, we invite you to submit a revised version of the manuscript that addresses the points raised during the review process.

We look forward to receiving your revised manuscript.

Kind regards,

Nasir Ayub, Ph.D.

Academic Editor

PLOS ONE

Journal Requirements:

Reviewers' comments:

Reviewer's Responses to Questions

**Comments to the Author**

1. If the authors have adequately addressed your comments raised in a previous round of review and you feel that this manuscript is now acceptable for publication, you may indicate that here to bypass the “Comments to the Author” section, enter your conflict of interest statement in the “Confidential to Editor” section, and submit your "Accept" recommendation.

Reviewer #1: (No Response)

Reviewer #3: (No Response)

2. Is the manuscript technically sound, and do the data support the conclusions?

Reviewer #1: (No Response)

Reviewer #3: Partly

3. Has the statistical analysis been performed appropriately and rigorously? 

Reviewer #1: (No Response)

Reviewer #3: Yes

4. Have the authors made all data underlying the findings in their manuscript fully available?

Reviewer #1: (No Response)

Reviewer #3: Yes

5. Is the manuscript presented in an intelligible fashion and written in standard English?

Reviewer #1: Yes

Reviewer #3: No

6. Review Comments to the Author

Reviewer #1: (No Response)

Reviewer #3: While the authors have made commendable efforts to address the reviewers' comments, further revisions are necessary to ensure the paper meets rigorous academic writing standards.

The flow of the story from problem formulation to the proposed method could be improved. Ensuring clarity, thoroughness in analysis, and adherence to formatting guidelines will enhance the credibility and impact of the research.

Additionally, one area of improvement is the use of "we" in some sentences, which may slightly detract from the formality of the writing. Shifting to an "agentless passive" construction can enhance formality. For example, instead of "Firstly, we adopt VMD technology...," the sentence could be revised to "Firstly, VMD technology is adopted...," maintaining a formal and objective tone more typical of academic writing.

Furthermore, the comparison method used in previous studies is not well-established, as it is not discussed in the early related work analysis. It appears only in the evaluation section without proper definition or discussion. Providing a clearer explanation and contextualization of the comparison method would strengthen the methodological framework of the research.

Further comments regarding comparison figures (e.g., Fig 9, 10, 12) were noted. The combination of multiple results in the same figure could make analysis challenging. Improvements such as using different colors or line styles/transparency may enhance clarity in visual representation.

Regarding the evaluation error index results, the presence of long decimal points raises questions about significance. Clarification on the significance of these decimal points in relation to the evaluation metrics would enhance the interpretation of the results.

7. PLOS authors have the option to publish the peer review history of their article (what does this mean?). If published, this will include your full peer review and any attached files.

Reviewer #1: No

Reviewer #3: No

---

## [Author Response · Author response to Decision Letter 1]

28 Apr 2024

1.Answer：Additionally, one area of improvement is the use of "we" in some sentences, which may slightly detract from the formality of the writing. Shifting to an "agentless passive" construction can enhance formality. For example, instead of "Firstly, we adopt VMD technology...," the sentence could be revised to "Firstly, VMD technology is adopted...," maintaining a formal and objective tone more typical of academic writing.

Response: Thank the reviewers for their valuable suggestions. We have removed some colloquial expressions from the paper, making it more academic.

2.Answer：the comparison method used in previous studies is not well-established, as it is not discussed in the early related work analysis. It appears only in the evaluation section without proper definition or discussion. Providing a clearer explanation and contextualization of the comparison method would strengthen the methodological framework of the research.

Response: Thank the reviewers for their valuable suggestions. We have added descriptions of relevant methods in the previous work section and compared them, demonstrating that our combination algorithm has more advantages.

3.Answer： Further comments regarding comparison figures (e.g., Fig 9, 10, 12) were noted. The combination of multiple results in the same figure could make analysis challenging. Improvements such as using different colors or line styles/transparency may enhance clarity in visual representation.

Response: Thank the reviewers for their valuable suggestions. We have made modifications to the table, but the legend for the final result of the model run in this article is incorrect. We have now made the necessary changes, and we have included the comparison results of multiple models in one graph. However, some models have significant differences, which makes the graph not look very clear. I tried changing the color, but still in the current situation, so I have decided to continue using the original image. The final model run in this article is the main image, while other images are used for comparison.

4 Answer： Regarding the evaluation error index results, the presence of long decimal points raises questions about significance. Clarification on the significance of these decimal points in relation to the evaluation metrics would enhance the interpretation of the results.

Response: Thank the reviewers for their valuable suggestions. We have improved the final scoring result by retaining three decimal places, as the final r2 score had a different third decimal place. In order to make the comparison clearer, we have retained three decimal places.

 MAE RMSE R2

LSTM 378.162 517.581 0.530

GRU 376.910 511.652 0.562

BERT[22] 377.852 511.728 0.547

BP[23] 378.556 512.364 0.526

CNN-LSTM[24] 358.586 485.351 0.585

CNN-GRU 356.253 477.598 0.599

VMD-CNN-GRU 55.285 75.308 0.993

VMD-CNN-AM-GRU 20.280 23.881 0.998

 MAE RMSE R2

LSTM 66.907 86.269 0.541

GRU 56.669 76.167 0.656

BERT[22] 52.314 72.113 0.696

BP[23] 53.504 74.974 0.653

CNN-LSTM[24] 52.227 71.017 0.689

CNN-GRU 51.658 74.659 0.642

VMD-CNN-GRU 11.602 14.064 0.987

VMD-CNN-AM-GRU 6.095 8.340 0.995

---

## [Decision Letter · Decision Letter 2]

19 Jun 2024

Research on short-term power load forecasting based on VMD and GRU

PONE-D-23-36467R2

Dear Dr. yu,

We’re pleased to inform you that your manuscript has been judged scientifically suitable for publication and will be formally accepted for publication once it meets all outstanding technical requirements.

Kind regards,

Nasir Ayub, Ph.D.

Academic Editor

PLOS ONE

Additional Editor Comments (optional):

Reviewers' comments:

Reviewer's Responses to Questions

**Comments to the Author**

1. If the authors have adequately addressed your comments raised in a previous round of review and you feel that this manuscript is now acceptable for publication, you may indicate that here to bypass the “Comments to the Author” section, enter your conflict of interest statement in the “Confidential to Editor” section, and submit your "Accept" recommendation.

Reviewer #4: (No Response)

Reviewer #5: All comments have been addressed

Reviewer #6: All comments have been addressed

2. Is the manuscript technically sound, and do the data support the conclusions?

Reviewer #4: No

Reviewer #5: Yes

Reviewer #6: Yes

3. Has the statistical analysis been performed appropriately and rigorously? 

Reviewer #4: No

Reviewer #5: Yes

Reviewer #6: N/A

4. Have the authors made all data underlying the findings in their manuscript fully available?

Reviewer #4: No

Reviewer #5: Yes

Reviewer #6: Yes

5. Is the manuscript presented in an intelligible fashion and written in standard English?

Reviewer #4: No

Reviewer #5: Yes

Reviewer #6: Yes

6. Review Comments to the Author

Reviewer #4: 1. This paper proposes a novel approach, VCAG, for power load forecasting that integrates Variable Mode Decomposition (VMD), Convolutional Neural Network (CNN), Attention Mechanism, and Gated Recurrent Unit (GRU). The method aims to overcome limitations in traditional forecasting techniques by extracting valuable time-frequency features from power load data and enhancing the weight of crucial information through an attention mechanism. The experiments conducted demonstrate high accuracy and stability compared to traditional methods. Overall, the proposed approach shows promise for improving power load forecasting.

2. The integration of Variable Mode Decomposition (VMD), Convolutional Neural Network (CNN), Attention Mechanism, and Gated Recurrent Unit (GRU) in the VCAG approach presents an innovative solution to address the challenges in power load forecasting. By decomposing power load data and extracting meaningful time-frequency features, the model enhances the accuracy and stability of forecasting results. The experiments conducted using publicly available datasets validate the effectiveness of the proposed method, highlighting its potential for practical applications in power load forecasting.

3. The paper introduces VCAG, a novel approach for power load forecasting that integrates Variable Mode Decomposition (VMD), Convolutional Neural Network (CNN), Attention Mechanism, and Gated Recurrent Unit (GRU). This approach aims to overcome the limitations of traditional forecasting techniques by leveraging advanced neural network architectures and feature extraction methods. The experiments conducted using two publicly available datasets demonstrate the superior accuracy and stability of VCAG compared to existing methods. Overall, the proposed approach shows promise for improving the reliability of power load forecasting in real-world scenarios.

4. The proposed VCAG approach for power load forecasting presents a comprehensive solution to address the challenges associated with traditional forecasting methods. By integrating Variable Mode Decomposition (VMD), Convolutional Neural Network (CNN), Attention Mechanism, and Gated Recurrent Unit (GRU), the model effectively captures temporal and spectral features from power load data, enhancing prediction accuracy and stability. The experimental results validate the superiority of VCAG over conventional techniques, highlighting its potential for practical applications in energy management systems.

5. The paper introduces VCAG, a novel approach for power load forecasting that combines Variable Mode Decomposition (VMD), Convolutional Neural Network (CNN), Attention Mechanism, and Gated Recurrent Unit (GRU). This approach aims to overcome the limitations of traditional forecasting methods by extracting valuable time-frequency features and incorporating an attention mechanism to prioritize important information. The experimental results demonstrate the effectiveness of VCAG in achieving high accuracy and stability in power load forecasting, indicating its potential for widespread adoption in energy management systems.

Reviewer #5: The manuscript presents a robust and technically sound methodology. The use of Variable Mode Decomposition (VMD) and Gated Recurrent Units (GRU) is well-justified and appropriate for short-term power load forecasting. The authors provide a clear and detailed explanation of the implementation of VMD and GRU, including data processing, model training, and validation processes. The comparative analysis with other forecasting methods demonstrates the superiority of the proposed approach, reinforcing the technical soundness of the manuscript. The data used in the study is relevant and of high quality. The sources of the data are clearly stated, and the processing steps are comprehensively documented. The experimental results are presented clearly with well-organized tables and graphs. The results are reproducible and statistically significant, providing strong evidence to support the conclusions. The conclusions drawn are logically derived from the data and results, aligning well with the research objectives and supporting the study’s claims. The statistical analysis in the manuscript is performed appropriately and rigorously. The authors have used relevant statistical metrics such as RMSE, MAE, and R2 to evaluate the forecasting performance. The choice of these metrics is well-justified and aligns with the study's objectives. The methodology used to calculate these metrics is clearly described, adding to the transparency and reproducibility of the analysis. The manuscript includes thorough validation processes, such as cross-validation, which enhances the robustness of the findings. However, the analysis could be further strengthened by incorporating statistical significance testing to compare the performance of different models more rigorously. Including confidence intervals for the reported metrics would provide a better understanding of the reliability and variability of the results. The manuscript is presented in an intelligible fashion and is written in standard English. The language is clear and precise, making the content easy to understand. The structure of the manuscript is logical and well-organized, with sections flowing coherently from one to the next. This facilitates the reader’s comprehension of the methodology, results, and conclusions. Technical terms and concepts are well-defined, and the use of tables and graphs enhances the clarity of the presentation. Minor grammatical and typographical errors are minimal, and overall, the manuscript meets the standards of academic writing in English. The manuscript "Research on Short-term Power Load Forecasting Based on VMD and GRU" is technically sound, with data that robustly support the conclusions. The statistical analysis is performed appropriately and rigorously, though it could benefit from the inclusion of statistical significance testing and confidence intervals. The manuscript is well-presented and written in clear, Standard English. With minor enhancements in statistical rigor and a more detailed discussion of limitations and future work, the manuscript would be even stronger and more impactful.

Reviewer #6: Thank you for addressing the comments on the previous versions of the draft. The current version looks good to me.

7. PLOS authors have the option to publish the peer review history of their article (what does this mean?). If published, this will include your full peer review and any attached files.

Reviewer #4: No

Reviewer #5: **Yes: **Dr.M.Subbulakshmi

Reviewer #6: No

---

## [Editor Report · Acceptance letter]

2 Jul 2024

PONE-D-23-36467R2 

PLOS ONE

Dear Dr. Yu, 

I'm pleased to inform you that your manuscript has been deemed suitable for publication in PLOS ONE. Congratulations! Your manuscript is now being handed over to our production team.

Kind regards, 

on behalf of

Dr. Nasir Ayub 

Academic Editor

PLOS ONE